# The RALE Score Versus the CT Severity Score in Invasively Ventilated COVID-19 Patients—A Retrospective Study Comparing Their Prognostic Capacities

**DOI:** 10.3390/diagnostics12092072

**Published:** 2022-08-26

**Authors:** Christel M. Valk, Claudio Zimatore, Guido Mazzinari, Charalampos Pierrakos, Chaisith Sivakorn, Jutamas Dechsanga, Salvatore Grasso, Ludo Beenen, Lieuwe D. J. Bos, Frederique Paulus, Marcus J. Schultz, Luigi Pisani

**Affiliations:** 1Department of Intensive Care & Laboratory of Experimental Intensive Care and Anesthesiology (L·E·I·C·A), Amsterdam UMC, Location ‘AMC’, 1105 AZ Amsterdam, The Netherlands; 2Department of Emergency and Organ Transplantation, University of Bari Aldo Moro, 70124 Bari, Italy; 3Department of Anaesthesiology and Critical Care, Hospital Universitario y Politecnico la Fe, 46026 Valencia, Spain; 4Perioperative Medicine Research Group, Instituto de Investigación Sanitaria la Fe, 46026 Valencia, Spain; 5Department of Intensive Care, Centre Hospitalier Universitaire Brussels, 1020 Brussels, Belgium; 6Intensive Care Unit, NHS Foundation Trust, University College London Hospitals, London NW1 2BU, UK; 7Division of Pulmonary and Critical Care, Department of Medicine, Chonburi Hospital, Chonburi 20000, Thailand; 8Department of Radiology, Amsterdam UMC, Location ‘AMC’, 1105 AZ Amsterdam, The Netherlands; 9Department of Pulmonology, Amsterdam UMC, Location ‘AMC’, 1105 AZ Amsterdam, The Netherlands; 10Mahidol-Oxford Tropical Medicine Research Unit (MORU), Mahidol University, Bangkok 10400, Thailand; 11Nuffield Department of Medicine, University of Oxford, Oxford OX1 2JD, UK; 12Anaesthesia and Intensive Care Unit, Miulli Regional Hospital, 70021 Acquaviva Delle Fonti, Italy

**Keywords:** ARDS, ICU, coronavirus disease 2019, mechanical ventilation, chest X-ray, chest computed tomography, chest imaging, RALE score, CT severity score, CTSS

## Abstract

Background: Quantitative radiological scores for the extent and severity of pulmonary infiltrates based on chest radiography (CXR) and computed tomography (CT) scan are increasingly used in critically ill invasively ventilated patients. This study aimed to determine and compare the prognostic capacity of the Radiographic Assessment of Lung Edema (RALE) score and the chest CT Severity Score (CTSS) in a cohort of invasively ventilated patients with acute respiratory distress syndrome (ARDS) due to COVID-19. Methods: Two-center retrospective observational study, including consecutive invasively ventilated COVID-19 patients. Trained scorers calculated the RALE score of first available CXR and the CTSS of the first available CT scan. The primary outcome was ICU mortality; secondary outcomes were duration of ventilation in survivors, length of stay in ICU, and hospital-, 28-, and 90-day mortality. Prognostic accuracy for ICU death was expressed using odds ratios and Area Under the Receiver Operating Characteristic curves (AUROC). Results: A total of 82 patients were enrolled. The median RALE score (22 [15–37] vs. 26 [20–39]; *p* = 0.34) and the median CTSS (18 [16–21] vs. 21 [18–23]; *p* = 0.022) were both lower in ICU survivors compared to ICU non-survivors, although only the difference in CTSS reached statistical significance. While no association was observed between ICU mortality and RALE score (OR 1.35 [95%CI 0.64–2.84]; *p* = 0.417; AUC 0.50 [0.44–0.56], this was noticed with the CTSS (OR, 2.31 [1.22–4.38]; *p* = 0.010) although with poor prognostic capacity (AUC 0.64 [0.57–0.69]). The correlation between the RALE score and CTSS was weak (r^2^ = 0.075; *p* = 0.012). Conclusions: Despite poor prognostic capacity, only CTSS was associated with ICU mortality in our cohort of COVID-19 patients.

## 1. Introduction

Quantitative scores for pulmonary infiltrates and consolidations on lung images such as chest radiography (CXR) and chest computed tomography (CT) are increasingly used in critically ill invasively ventilated patients. Such visual scores may have comparable diagnostic and prognostic capacities [1,2,3]. Advantages of a CXR-based score over chest CT scan-based score rely on CXRs being easier and cheaper to obtain, with no patient transportation outside of the ICU, and lower radiation exposure than a chest CT scan.

The Radiographic Assessment of Lung Edema (RALE) score quantifies both extent and severity of parenchymal abnormalities in the CXR [4]. The RALE score has been shown to have an excellent diagnostic accuracy for acute respiratory distress syndrome (ARDS) [5,6,7], and early changes in the RALE score have been found to have an association with outcome in critically ill invasively ventilated patients [4,8,9]. This was confirmed in invasively ventilated patients with coronavirus disease 2019 (COVID-19) [1]. A recent analysis on invasively ventilated patients showed how higher RALE scores were associated with worse parameters of respiratory mechanics and higher levels of pro-inflammatory biomarkers [10]. The CT severity score (CTSS) quantifies the extent of parenchymal involvement on the chest CT scan, by summing the percentage of affected lung tissue per each lung lobe [11]. The CTSS has been used successfully in the diagnostic process in COVID-19 patients and has been shown to correlate well with disease severity and laboratory parameters [12]. The CTSS may even have a correlation with short-term outcome [13,14].

In the current study, we aimed to determine the association with mortality of the RALE score and the CTSS in critically ill invasively ventilated COVID-19 patients and compare their prognostic capacity for ICU mortality. We also sought to determine the correlation between the RALE score and the CTSS. We hypothesized that the RALE score and the CTSS have comparable prognostic capacity for outcome.

## 2. Materials and Methods

### 2.1. Study Design

This was a retrospective observational study in critically ill invasively ventilated COVID-19 patients admitted to the intensive care units (ICUs) of two tertiary centers, the Academic Medical Center, and the Free University Medical Center, Amsterdam, The Netherlands. The Institutional Review Boards of both hospitals approved the study protocol (approval W20_494 # 20.546). The need for individual patient informed consent was waived because the data used for this analysis were collected as part of standard care. The study is registered at clinicaltrials.gov (NCT05047653).

### 2.2. Inclusion and Exclusion Criteria

Consecutive patients admitted between 1 March 2020, and 1 June 2020, the first wave of the national outbreak in the Netherlands, and between 1 October 2020 and 31 December 2020, the first 3 months of the second wave of the national outbreak in the Netherlands were screened for participation. Patients were selected if having received invasive ventilation for acute hypoxemic respiratory failure due to COVID-19 that was confirmed by reverse transcriptase–polymerase chain reaction for SARS-CoV-2. Patients aged <18 years, patients with an alternate diagnosis, and patients receiving other forms of oxygen support, such as high flow nasal oxygen (HFNO), noninvasive ventilation, or continuous positive airway pressure, were excluded. We also excluded patients that had their first CXR and first chest CT scan too far apart in time, using a cutoff of 24 h.

### 2.3. Data Collection

An online case report form created with Castor (www.castoredc.com (accessed on 29 July 2022)) was used to collect baseline and demographic characteristics, the Acute Physiology and Chronic Health Evaluation (APACHE) II score, and typical ventilation parameters at the moment of lung imaging, including inhaled oxygen fraction (FiO_2_), positive end-expiratory pressure (PEEP), maximum airway pressure (Pmax), respiratory rate (RR), tidal volume (VT), and the blood gas analysis results.

### 2.4. Imaging Scores

CXRs and chest CT scans were collected from the electronic imaging systems in each hospital and uploaded in Joint Photographic Experts Group (JPEG) format into the database. Then, CXRs were scored by at least two independent scorers that were extensively trained in calculating RALE scores. For this, each scorer was trained in the RALE scoring by one of the investigators (C.Z.), who was trained during a 1-month focused period by the team that developed the RALE score [6]. An interclass correlation coefficient (ICC) > 0.8 between the trainer and other scorers on a training sample of 22 CXRs from another set of CXRs of patients with ARDS was a prerequisite for scoring CXRs in the study dataset. A third scorer was only involved if the difference in numeric RALE score between two scorers was >25%, in order to reach a final consensus by discussion. Chest CT scans were scored by a radiologist experienced in chest CT.

For the RALE score, the chest was divided into four quadrants by a vertical line over the spine and a horizontal line at the level of the first branch of the left main bronchus; each quadrant was then scored for extent of alveolar opacities (consolidation score, from 0 to 4), and the corresponding density of alveolar opacities (density score, from 1 to 3), and the final score was the sum of the product of the consolidation and density scores for each quadrant. The RALE score thus ranged from 0 (no abnormalities) to 48 (maximum abnormalities) [4]. For details on RALE score computation, see Appendix A.

For the CTSS, the percentage of involvement per each lung lobe was scored and summed; the final score was the sum of the individual lobar scores, which could range from 0 (no lung involvement) to 25 (maximum involvement when all the 5 lobes show more than 75% involvement) [15]. For details, see Appendix A.

### 2.5. Outcomes

The primary outcome was ICU mortality. Secondary outcomes were duration of ventilation in survivors, length of stay in ICU, and hospital-, 28-, and 90-day mortality.

### 2.6. Power Calculation

We did not perform a formal sample size calculation. Instead, the available patients served as the sample size for this study.

### 2.7. Statistical Analysis

Demographic data and clinical and outcome variables were summarized as medians (interquartile range) for continuous variables and as frequencies (percentage) for categorical variables. Normally distributed variables were compared between groups with *t*-test or ANOVA. Not normally distributed variables were compared between groups with Wilcoxon signed rank tests or Mann–Whitney U test. Categorical variables were compared between groups by Wilcoxon signed rank test.

To test the association of the radiological scores with outcomes, we performed univariable and multivariable logistic regression models introducing the RALE score and the CTSS alternatively. As covariates for the logistic regression models, we used age, body mass index, and the APACHE II score [16]. We performed a sensitivity analysis introducing PEEP as covariate, as PEEP can influence imaging scores. In a further sensitivity analysis, we used a Splines fitted model to take into consideration the non-normal distribution of the CTSS. For this purpose, linear tail restricted cubic splines with three knots and three degrees of freedom were fitted, as by default values from the ‘rcs’ function of the ‘rms’ R package. For both univariable and multivariable logistic regression model, a receiver operating characteristic (ROC) was constructed from which the area under curve (AUROC) was calculated. In order to assess the prognostic potential of both scores, ROC curves were estimated on the averaged terms after applying a repeated (5 times) 10-fold cross-validation algorithm to the dataset, and 95% confidence interval on ROC estimates were obtained by 500 bootstrap repetitions. When the AUROC was 0.9–1.0, the prognostic capacity was considered excellent, if 0.8–0.9, 0.7–0.8, and 0.6–0.7, the test was defined as good, fair, or poor, respectively. The De Long test was used for the comparison of the AUROCs [17]. A non-significant result would confirm that the two scores are comparable in terms of prognostic capability for ICU mortality. For the correlation between the RALE and CTSS, we used the coefficient of determination obtained from a simple linear regression model (r^2^), using RALE as independent variable and CTSS as dependent variable.

To assess the continuous outcomes such as duration of ventilation and ICU length of stay in survivors, we fitted a linear regression model for each score with the same covariates structure as the aforementioned logistic models. We compared which score performs best in the model using R^2^.

All analyses were performed using a two-sided superiority hypothesis test, with a significance level of 0.05 and presented with two-sided 95% confidence intervals. No corrections were performed for multiple comparisons across secondary clinical outcomes, thus the findings should be considered as exploratory. All analyses were performed using R (version 4.0.2, R Core Team, 2016, Vienna, Austria).

## 3. Results

### 3.1. Patients

From 1 March 2020 through 1 June 2020 (the first 3 months of the first wave) and from 1 October 2020 through 31 December 2020 (the first 3 months of the second wave), we screened 254 patients (Figure 1). We excluded 172 patients for various reasons, the most frequent one being a missing CXR or CT scan within the predefined timespan of 24 h.

Patient demography and ventilation characteristics and outcomes are shown in Table 1 and Table 2. The median age was 65 [60–72] years, with the most common comorbidities being hypertension and diabetes. There were no significant differences in frequencies in demographic characteristics and comorbidities between ICU survivors and ICU non-survivors except for hypertension. Furthermore, ventilation parameters were not different between the same groups and most patients had moderate or severe ARDS. ICU mortality was 42.7%.

### 3.2. Imaging Scores

The median RALE score was 22 [15–37] in ICU survivors and 26 [19–39] in non–ICU survivors (*p* = 0.34); the median CTSS was lower in ICU survivors compared to ICU non-survivors (18 [16–21] vs. 21 [18–23]; *p* = 0.02). The CTSS showed a skewed distribution towards higher values (Figure 2). The correlation between the two scores was poor, with very little variance in RALE being explained by the CTSS (r^2^ = 0.075; *p* = 0.012) (Figure 3). In 19.4% of CXRs, a third scorer was needed to reach a consensus regarding the RALE score.

### 3.3. Prognostic Capacity for ICU Death

The RALE score had no association with ICU mortality (OR, 1.35 [95%CI 0.64–2.84]; *p* = 0.42) (Table 3), neither showed prognostic capacity, with an area under ROC (AUROC) for ICU mortality of 0.50 [0.44–0.56] (Figure 4), although the calibration of the fitted model was poor (Appendix A). The CTSS had an association with ICU mortality (OR, 2.31 [95%CI 1.22–4.38]; *p* = 0.01), with an adequate calibration of the fitted model (Appendix A). The prognostic capacity of the CTSS was poor, with an AUROC for ICU mortality of 0.64 [0.57–0.69] (Figure 4), yet significantly superior to the RALE score (*p* value for De Long test = 0.006) (Table 4).

The CTSS had an association with hospital mortality, 28-day, and 90-day mortality, but not with length of stay or duration of ventilation (Table 3). The RALE score was not associated with any of the secondary outcomes.

### 3.4. Sensitivity Analyses

The addition of PEEP in the model and using splines fitted models did not modify the findings for the primary outcome (Appendix A). Both scores were not associated with ICU mortality when only the CXRs of the first 3 days were taken into consideration (Table 3).

## 4. Discussion

The main findings of this study in COVID-19 patients with ARDS can be summarized as follows: (1) The first available CTSS score is associated with patient mortality albeit with a poor prognostic capacity; (2) the first available RALE score has no association with outcome and no prognostic capacity; (3) there is a lack of correlation between the two radiological scores; (4) none of the scores can predict ICU length of stay or duration of mechanical ventilation.

In contrast to the study hypothesis, the two radiological scores showed a different behavior with regards to prognostication of patient outcome. The CT-based score was significantly higher in non-survivors and showed an association with all mortality outcomes. Although the CTSS was initially developed to discriminate the severity of disease [15,18], subsequent studies did show a consistent association with outcome. In a validation study performed in the Netherlands, the CTSS was associated with 30-day mortality, although that study was performed in the emergency department and recruited suspected rather than confirmed COVID-19 cases [14]. In ICU patients, the CTSS was shown to predict the composite outcome of death or ICU stay for more than 30 days [19].

Despite showing an association with mortality and a superior predictive potential as compared to the RALE score, the prognostic capability of the CTSS was minimal and unfit for clinical purposes. This highlights the complexity of the trajectory of COVID-19 ARDS, where the extent of pulmonary impairment as assessed by imaging at baseline has a limited impact on survival. The poor prognostic capacity of the CTSS was also observed in the emergency department [14]. Although chest CT manages to identify progression of COVID-19 ground-glass opacities towards consolidations and absorption [20], the baseline score seems to provide scarce predictive information. The failure to predict mortality using baseline scores has also been shown for lung ultrasound [21] and chest X-ray [1]. The use of early changes in radiological scores seem to outperform the use of baseline values for prognostication purposes [1,8,14].

The RALE score had consistently no association with any of the mortality outcomes. This echoes the finding of a recent international multicenter study performed on 139 patients with COVID-19 ARDS [1] and another study in five German ICUs [22]. Yet, these findings are in contrast with other studies conducted in less severe cohorts or patients outside of the ICU that did find an association between the entity of pulmonary impairment estimated by the RALE score and adverse outcomes [8,23,24,25]. The RALE score provides a reliable interpretation of signs of lung edema on chest radiographs and has been validated as a good predictor of ARDS [4,8]. The RALE may also be associated with long term diffusion impairment, as recently shown in a cohort of patients performing CXR at six months from discharge [26]. COVID-19 worked as a boost for a development of machine learning solutions to assist specialists in early diagnostic detection and treatment of ICU patients [27]. For instance, the RALE score was also used as benchmark to validate a fully automated segmentation and intensity quantification method in CXRs for COVID–19 patients [28]. To date, the overall evidence is against the routine use of the baseline RALE score for the prediction of outcome in mechanically ventilated ICU patients with COVID-19 ARDS.

Pulmonary vascular dysfunctions described in COVID-19 ARDS are not captured by the CTSS and RALE score [29,30,31,32,33,34]. Aside from thrombo-embolic complications, mortality may be also driven by other factors, such as bacterial and fungal infections complications [35,36] and ICU-acquired weakness. The initial ventilator management was also shown to moderate the prognostic value of other predictive parameters [37]. These mechanisms aid to justify the absent or scarce prognostic potential of baseline radiological scores in COVID-19 ARDS.

The negligible correlation between CXR RALE score and the chest CT scan CTSS was unexpected and does not have a univocal explanation [38,39]. In fact, several conceptual differences may impede interchangeable use of the two scores. The RALE attempts to quantify both extent and severity of alveolar infiltrates of pulmonary edema while the CTSS solely estimates the extent of lung involvement [13,14,15,18]. Secondly, the different distributions observed suggest that caution should be taken before comparing a CXR based score with a CT derived one. For instance, the CTSS was skewed towards higher values, while the RALE showed a normal distribution. For the bedside clinician, chest radiographs remain easier to perform, require lower radiation dose, and are safer compared to CT. The possible benefits of routine CT examinations (fast triage, high resolution imaging, association with mortality) may not outweigh the harms such as overuse of medical resources and higher radiation dose [14]. Considered these shortcomings and the poor prognostic capacity of the CTSS, the use of early changes in the RALE score or lung ultrasound score could be more useful as first line imaging technique [1,8,19,21].

The study was designed to minimize bias by strictly adhering to a predefined statistical analysis plan and systematic training of the RALE scorers. We had a low interobserver variability between the scorers [6,7,9,40]. We collected data in both the first as the second wave in the Netherlands, with minimal loss to follow-up.

This study has several limitations. First, the sample size of this study was relatively small in combination with a high overall mortality and the retrospective design of this study limits the inclusion of all potential confounders. Secondly, the inclusion criteria of the study could have resulted in selection bias and patient-level differences in terms of treatments received. This study only included mechanical ventilated patients with both a CT scan and CXR within the same timeframe and ICU length of stay for at least 24 h. This may potentially create a systematic bias towards patients with higher severity or generate a risk of attrition bias. Finally, despite the high ICC among scorers, there was still significant variability in one out of five CXRs with the need of a third scorer. This additional scorer was not blinded to the results of the previous assessments, and this could have generated scoring bias.

## 5. Conclusions

In this cohort of invasively ventilated patients with ARDS due to COVID-19, the CTSS of the first available chest CT scan was associated with short- and medium-term mortality outcomes, albeit with poor prognostic capacity. The CXR-based RALE score was not associated with any mortality outcome and had no prognostic capacity. No correlation was found between the CXR and CT-based score, a finding to be validated in future studies. Neither score could predict duration of ventilation or length of stay in ICU.

## Figures and Tables

**Figure 1 diagnostics-12-02072-f001:**
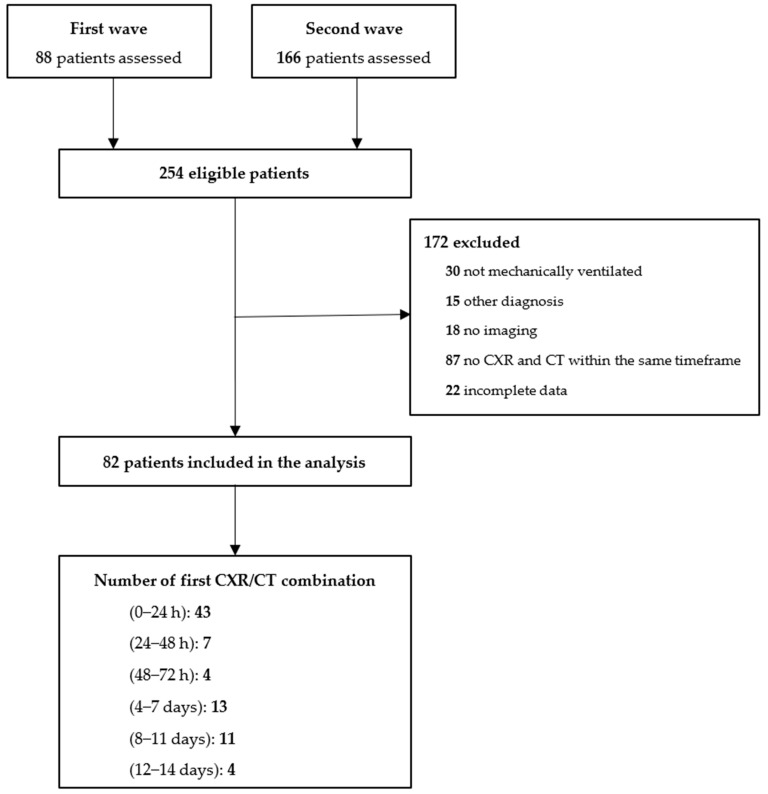
Patient flow. Consecutive patients were screened for participation. Patients were selected if having received invasive ventilation for acute hypoxemic respiratory failure due to COVID-19 that was confirmed by reverse transcriptase–polymerase chain reaction for SARS-CoV-2. Abbreviations: CXR: chest X-ray; CT: chest CT scan.

**Figure 2 diagnostics-12-02072-f002:**
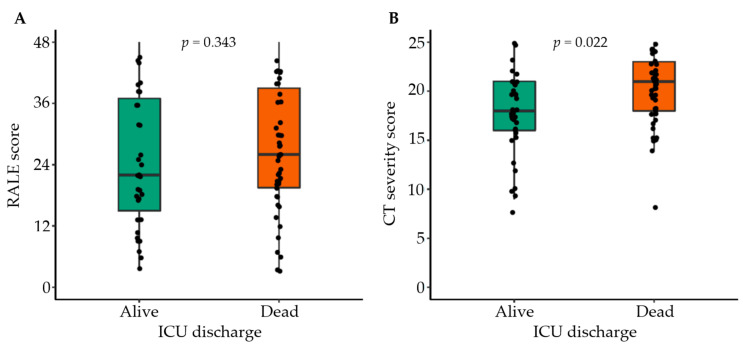
Boxplots of the (**A**) RALE score and (**B**) CTSS in survivors versus non-survivors.

**Figure 3 diagnostics-12-02072-f003:**
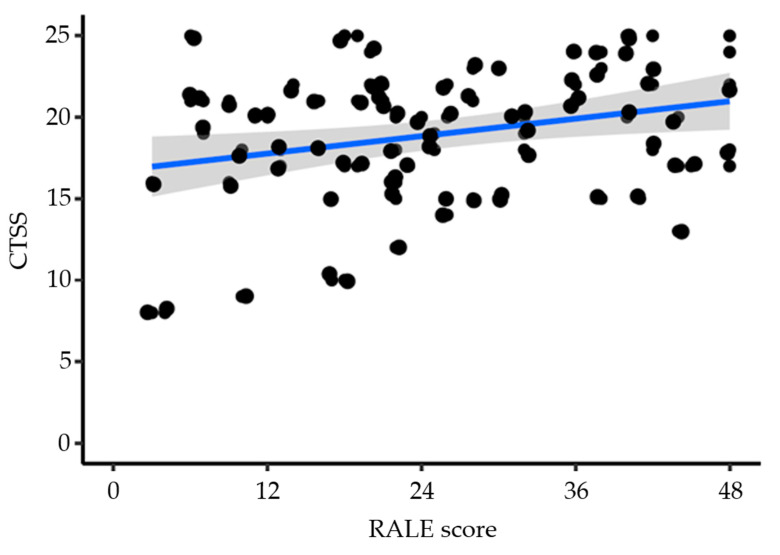
Scatterplot showing the correlation between the RALE score and the CTSS, with the linear regression estimation (in blue), and the 95%CI shown in shaded area. R^2^ = 0.075; *p* = 0.012. Beta coefficient for RALE score: 0.08 [95%CI 0.02–0.16].

**Figure 4 diagnostics-12-02072-f004:**
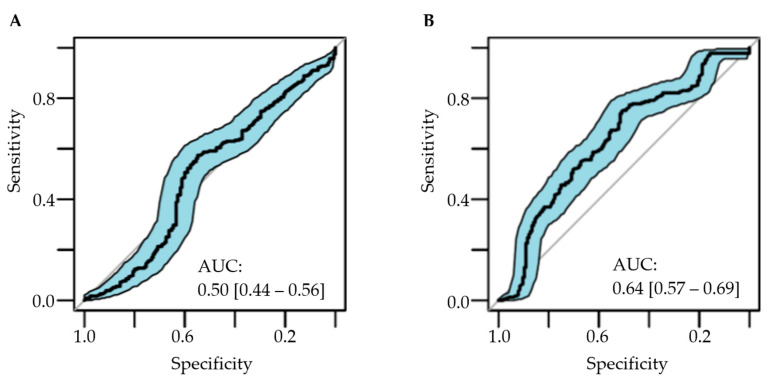
Discriminative capacity of (**A**) the RALE score and (**B**) the CTSS for ICU mortality. Shaded areas show 95% confidence intervals. Abbreviations: AUC: area under the curve; RALE: Radiographic Assessment of Lung Edema; CTSS: Chest CT Severity Score.

**Table 1 diagnostics-12-02072-t001:** Baseline characteristics of the patients.

	Overall *n* = 82	Alive * *n* = 35	Dead *n* = 47	*p*-Value	SMD
**demographics**					
Age, years (median [IQR])	65 [60–72]	65 [59–72]	65 [60–72]	0.669	0.174
Male gender, *n* (%)	60 (73.2)	24 (68.6)	36 (76.6)	0.576	0.181
Body mass index, kg∙m^2^ (median [IQR])					
**severity and comorbidities**					
APACHE II (median [IQR])	12.0 [10.0–20.0]	12.0 [10.0–15.5]	13.0 [10.0–20.0]	0.289	0.245
Comorbidities, yes, *n* (%)	46 (56.1)	20 (57.1)	26 (55.3)	1.000	
Arterial hypertension, *n* (%)	41 (50.0)	24 (68.6)	17 (36.2)	0.007	
Heart failure, *n* (%)	4 (4.9)	2 (5.7)	2 (4.3)	1.000	
Diabetes mellitus, *n* (%)	32 (39.0)	15 (42.9)	17 (36.2)	0.700	
Chronic kidney disease, *n* (%)	15 (18.3)	5 (14.3)	10 (21.3)	0.602	
Liver cirrhosis, *n* (%)	0 (0.0)	0 (0.0)	0 (0.0)	NA	
Chronic obstructive pulmonary disease, *n* (%)	5 (6.1)	3 (8.6)	2 (4.3)	0.733	
Active hematological malignancy, *n* (%)	4 (4.9)	1 (2.9)	3 (6.4)	0.830	
Active solid tumor malignancy, *n* (%)	3 (3.7)	2 (5.7)	1 (2.1)	0.794	
Metastatic cancer, *n* (%)	1 (1.2)	0 (0.0)	1 (2.1)	1.000	
Neuromuscular disease, *n* (%)	2 (2.4)	1 (2.9)	1 (2.1)	1.000	
Immunosuppression, *n* (%)	1 (1.2)	0 (0.0)	1 (2.1)	1.000	
Cardiovascular disease, *n* (%)	14 (17.1)	6 (17.1)	8 (17.0)	1.000	
AIDS, *n* (%)	0 (0.0)	0 (0.0)	0 (0.0)	NA	
Asthma, *n* (%)	3 (3.7)	2 (5.7)	1 (2.1)	0.794	
Obstructive sleep apnea, *n* (%)	2 (2.4)	0 (0.0)	2 (4.3)	0.609	
Other, *n* (%)	35 (42.7)	16 (45.7)	19 (40.4)	0.800	
**outcomes**					
Successful extubation, *n* (%)	35 (43.2)	33 (97.1)	2 (4.3)	<0.001	
RALE score (median [IQR])	25 [18–38]	22 [15–37]	26 [19–39]	0.343	0.187
CT severity score (median [IQR])	20 [17–22]	18 [16–21]	21 [18–23]	0.022	0.545
Intubation time-days (median [IQR])	10 [6–28]	9 [5–23]	40 [36–43]	0.117	1.638
ICU length of stay-days (median [IQR])	12 [7–26]	12 [7–26]	-	-	-
Hospital length of stay-days (median [IQR])	28 [16–53]	28 [16–53]	-	-	-

Data are median (quartile 25–75%), mean (±SD) or *n* (%). Percentages may not total 100 because of rounding. Abbreviations: ICU: intensive care unit; SMD: standardized mean difference; IQR: interquartile range; CT: computed tomography; RALE: Radiographic Assessment of Lung Edema. * At day ICU discharge.

**Table 2 diagnostics-12-02072-t002:** Ventilation parameters measured with the first available combination of CXR and CT imaging.

	Overall	Alive *	Dead	*p*-Value	SMD
	*n* = 82	*n* = 35	*n* = 47		
Ventilation mode, *n* (%)				0.587	0.326
Pressure controlled	26 (33.3)	8 (25.0)	18 (39.1)		
Pressure support	12 (15.4)	6 (18.8)	6 (13.0)		
Volume controlled	2 (2.5)	1 (2.9)	1 (2.1)		
ASV/Intellivent	10 (12.8)	5 (15.6)	5 (10.9)		
Spontaneous	30 (38.5)	13 (40.6)	17 (37.0)		
PEEP, cmH_2_O (median [IQR])	10 [9–12]	10 [9–12]	10 [8–12]	0.969	0.035
FiO_2_, % (median [IQR])	65 [50–80]	61 [50–80]	70 [50–85]	0.368	0.234
Tidal expiratory volume set, ml (median [IQR])	458 [379–518]	435 [356–508]	466 [399–582]	0.155	0.381
Respiratory rate, breaths/min (median [IQR])	24 [20–29]	25 [21–29]	21.30 [19–28]	0.122	0.337
Peak pressure, cmH_2_O (median [IQR])	24 [19–31]	24 [20–31]	23 [19–29]	0.509	0.201
SpO_2_, % (median [IQR])	92 [90–94]	93 [91–94]	92 [90–95]	0.856	0.025
etCO_2_, kPa (median [IQR])	5.2 [4.2–6.1]	5.1 [3.9–5.6]	5.4 [4.4–6.2]	0.159	0.175
pH (median [IQR])	7.38 [7.32–7.44]	7.40 [7.33–7.45]	7.37 [7.31–7.42]	0.358	0.251
PaO_2_, kpa (median [IQR])	9.5 [8.3–10.5]	9.6 [8.2–10.6]	9.3 [8.4–10.5]	0.914	0.125
PaO_2_/FiO_2_ (median [IQR])	113 [88–151]	119 [92–154]	109 [86–142]	0.583	0.149
PaCO_2_, kpa (median [IQR])	6.1 [5.3–7.6]	5.7 [5.3–7.4]	6.2 [5.5–7.9]	0.381	0.218

Data are median (quartile 25–75%) or *n* (%). Percentages may not total 100 because of rounding. Abbreviations: SMD: standardized mean difference; PEEP: positive end expiratory pressure; ASV: adaptive support ventilation; FiO_2_: fraction of inspired oxygen; PaO_2_: partial pressure of oxygen; etCO_2_: end-tidal carbon dioxide; PaCO_2_: partial pressure of carbon dioxide; IQR: interquartile range. * At day ICU discharge.

**Table 3 diagnostics-12-02072-t003:** Association between imaging scores and outcomes.

	RALE Score		CTSS	
Outcome	OR [95%CI] Univariable Estimate	OR [95%CI] Adjusted Estimate	OR [95%CI] Univariable Estimate	OR [95%CI] Adjusted Estimate
**Primary**				
ICU mortality	1.34 [0.67–2.69]	1.35 [0.64–2.84]	1.99 [1.11–3.54]	2.31 [1.22–4.38]
**Sensitivity analyses**				
Splines fitted model	2.97 [0.75–11.7]	1.67 [0.41–6.84]	2.98 [0.75–11.78]	2.31 [1.22–4.38]
Model with added PEEP		1.89 [0.81–4.38]		1.99 [1.025–3.87]
Only CXRs of the first 3 days	0.89 [0.39–2.03]	0.92 [0.39–2.16]	1.82 [0.91–3.63]	2.23 [0.99–4.98]
**Secondary**				
28-day mortality	1.04 [0.52–2.05]	1.02 [0.50–2.10]	1.79 [1.01–3.16]	1.96 [1.07–3.58]
Hospital mortality	1.21 [0.60–2.44]	1.19 [0.57–2.50]	1.70 [0.97–2.97]	1.94 [1.06–3.58]
90-day mortality	1.13 [0.58–2.26]	1.10 [0.53–2.29]	1.76 [1.01–3.10]	1.98 [1.10–3.65]
		Adjusted β linear coefficient [95%CI]		Adjusted β linear coefficient [95%CI]
Duration of ventilation in survivors	-	0.40 [−0.04–0.86] *p* = 0.075 ^$^	-	1.57 [0.10–3.03] *p* = 0.03 ^$^
ICU length of stay in survivors	-	0.30 [−0.20–0.81] *p* = 0.232 ^$^	-	1.48 [−0.12–3.09] *p* = 0.070 ^$^

Data are median (quartile 25–75%) or *n* (%). Percentages may not total 100 because of rounding. Table effect estimates from the fitted models. Abbreviations: RALE score: Radiographic Assessment of Lung Edema score; CTSS: Chest CT Severity Score; ICU: intensive care unit; OR: odds ratio; CI: confidence interval. ^$^
*p*-values for the beta coefficients of the linear regression output.

**Table 4 diagnostics-12-02072-t004:** Comparison of prognostic capacity across primary and secondary outcomes between the two imaging scores.

Outcome	AUC of Univariate ROC for RALE	AUC of Univariate ROC for CTSS	Comparison between AUCs(De Long Test)	AUC of Adjusted Logistic Regression Model for RALE *	AUC of Adjusted Logistic Regression Model for CTSS *	Comparison between AUCs (De Long)	R^2^ from the Adjusted Linear Model with RALE	R^2^ from the Adjusted Linear Model with CTSS
**Primary**								
ICU mortality	0.50 [0.44–0.56]	0.64 [0.57–0.69]	*p* = 0.001	0.55 [0.43–0.56]	0.66 [0.60–0.72]	*p* = 0.006	–	–
**Sensitivity analyses**								
Splines fitted model	0.53 [0.47–0.58]	0.56 [0.50–0.62]	*p* = 0.411	0.54 [0.47–0.58]	0.62 [0.57–0.68]	*p* = 0.005		
Only 3 days after ICU admission	0.66 [0.59–0.72]	0.60 [0.54–0.67]	*p* = 0.283	0.60 [0.49–0.61]	0.61 [0.60–0.72	*p* = 0.776		
**Secondary**								
28-day mortality	0.66 [0.61–0.71]	0.64 [0.58–0.69]	*p* = 0.445	0.53 [0.48–0.59]	0.65 [0.60–0.70]	*p* = 0.004	–	–
Hospital mortality	0.58 [0.53–0.64]	0.61 [0.55–0.61]	*p* = 0.507	0.48 [0.42–0.54]	0.64 [0.58–0.64]	*p* < 0.001	–	–
90-day mortality	0.57 [0.52–0.63]	0.63 [0.54–0.65]	*p* = 0.258	0.51 [0.45–0.57]	0.64 [0.59–0.70]	*p* < 0.001	–	–
Duration of ventilation in survivors	–	–	–	–	–	–	0.136	0.186
ICU length of stay in survivors	–	–	–	–	–	–	0.066	0.034

Data are median (quartile 25–75%) or *n* (%). Abbreviations: AUC: area under the curve; RALE: Radiographic Assessment of Lung Edema; CTSS: Chest CT Severity Score; ICU: intensive care unit; ROC: receiver operating characteristic. * Comparison between areas under the curve was tested using the De Long test.

## Data Availability

Requests for the data should be sent to Claudio Zimatore; email address: claudiozimatore@gmail.com.

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
