# Peer review of "The RALE Score Versus the CT Severity Score in Invasively Ventilated COVID-19 Patients—A Retrospective Study Comparing Their Prognostic Capacities"

_diagnostics, 2022, doi:10.3390/diagnostics12092072_

Round 1

Reviewer 1 Report

The authors have done an important research and presented in a well defined way with statistical analysis. However, the following comments/suggestions must be addressed before the publication. 

1. Line no. 60: change RALE core to Score

2. Line no. 70 "The aim of this current study is two fold" is giving a different impression, I suggest to rephrase the sentences.

3. Additionally, I believe the introductory section should be more descriptive especially about the RALE score. Moreover, if possible the recent studies must be incorporated while improving the introduction. The following studies can be utilized. 

  Safont B, Tarraso J, Rodriguez-Borja E, Fernández-Fabrellas E, Sancho-Chust JN, Molina V, Lopez-Ramirez C, Lope-Martinez A, Cabanes L, Andreu AL, Herrera S, Lahosa C, Ros JA, Rodriguez-Hermosa JL, Soriano JB, Moret-Tatay I, Carbonell-Asins JA, Mulet A, Signes-Costa J. Lung Function, Radiological Findings and Biomarkers of Fibrogenesis in a Cohort of COVID-19 Patients Six Months After Hospital Discharge. Arch Bronconeumol. 2022 Feb;58(2):142-149. doi: 10.1016/j.arbres.2021.08.014. Epub 2021 Sep 3. PMID: 34497426; PMCID: PMC8414844.

Al-Yousif N, Komanduri S, Qurashi H, Korzhuk A, Lawal HO, Abourizk N, Schaefer C, Mitchell KJ, Dietz CM, Hughes EK, Brandt CS, Fitzgerald GM, Joyce R, Chaudhry AS, Kotok D, Rivera JD, Kim AI, Shettigar S, Lavina A, Girard CE, Gillenwater SR, Hadeh A, Bain W, Shah FA, Bittner M, Lu M, Prendergast N, Evankovich J, Golubykh K, Ramesh N, Jacobs JJ, Kessinger C, Methé B, Lee JS, Morris A, McVerry BJ, Kitsios GD. Radiographic Assessment of Lung Edema (RALE) Scores are Highly Reproducible and Prognostic of Clinical Outcomes for Inpatients with COVID-19. medRxiv [Preprint]. 2022 Jun 14:2022.06.10.22276249. doi: 10.1101/2022.06.10.22276249. PMID: 35734089; PMCID: PMC9216727.

Rabaan AA, Tirupathi R, Sule AA, Aldali J, Mutair AA, Alhumaid S, Muzaheed, Gupta N, Koritala T, Adhikari R, Bilal M, Dhawan M, Tiwari R, Mitra S, Emran TB, Dhama K. Viral Dynamics and Real-Time RT-PCR Ct Values Correlation with Disease Severity in COVID-19. Diagnostics (Basel). 2021 Jun 15;11(6):1091. doi: 10.3390/diagnostics11061091. PMID: 34203738; PMCID: PMC8232180.

4. Line no. 92: change to 24 hours 

5. Line no. 181: The figure legend must be changed to explain the methodology to select the patients.  

6. Line no. 196: Carbon dioxide to carbon dioxide 

7. Line no. 253: patients with COVID–19 ARDS change it into COVID-19 patients with ARDS for clarity. 

8. In the conclusions authors mentioned "No correlation was found between the CXR and CT based score, and both scores could not predict duration of ventilation or length of stay in ICU"

However, there must be reasons and limitations should be provided which can suggest why there is no correlation

I wonder, if large scale studies with significantly higher number of patients will be conducted, there can be deviation from the present conclusion. 

Hence, I believe the conclusion must be improved while giving future directions to the scientific community.  

Author Response

Thank you.

Reviewer 2 Report

Thank you for giving me the opportunity to review this manuscript. In this small retrospective cohort study, the authors assessed the predictive accuracy of CXR and CT scan-based scores among patients hospitalized for COVID-19 with respect to ICU mortality, short-term mortality, and length of hospital stay. In summary, only CTSS was associated with ICU mortality, 28- and 90-day mortality, despite its poor prognostic ability. The manuscript is well written in all its parts, I have only some minor concerns:

-Abstract, results: briefly report significant associations of CTSS with secondary outcomes.

-Methods: clarify which tuning parameters you have set for realization of spline curves (number of knots, degrees of freedom, measure of spline curve centrality). Also report the spline graphical association of the 2 scores vs primary and secondary outcomes as Supplementary Figures.

-Methods: did you have any additional information on prescribed medications which might also have affected the association between scores and outcomes? if no, report this in study limitations. 

- Results, Line 172: a brief description of patients excluded because of missing data followed by a comparison with patients included in the study population should be added to study results; moreover, assessment of attrition bias should be performed by investigating the age and sex-adjusted association of the 2 scores with loss to follow-up.

-Results, Lines 174-175: briefly report the main features significantly characterizing patients who died vs patients who survived.

-Table 1-2: consistently use either No or N (and not both) for number of patients.

-Table 3: add value to beta coefficients of linear regression models.

3) Minor changes:

-Line 60, page 2: change "core" with "score".

-Line 202: remove one "in". 

Author Response

Thank you.

Reviewer 3 Report

I read with great interest the retrospective analysis of Valk et al. comparing the prognostic value of RALE- and CTSS-score in ventilated COVID-19 patients. I congratulate the authors to their work since the scientific structure and statistical analysis are very clearly prescribed. The limitations are also well explained. However, it should be noted that this obviously is a secondary analysis of pre-existing data, some of whose results have already been published elsewhere. Despite correct citation, I would advise the authors to state that in their Materials and Methods. The main scientific statement lies in comparison of the RALE-Score with the CTSS, where no correlation could be found, which indeed is an interesting fact. The conclusions drawn in the last paragraph on page 11 are of high clinical importance and could maybe pointed out more in the last conclusion paragraph.

Author Response

Thank you.

Reviewer 4 Report

Dear Authors

Many thanks for the opportunity to read this excellent work

The topic is very relevant, and you have correctly conducted the research. Machine learning is the future of multiple regression analysis and allows the development of a custom model to predict events.

In the introduction, if you want, let me suggest this reference (you are not obliged https://doi.org/10.3390/ijerph18147648; I'm not the author)

methods are excellent.

results and graphs are expressed in a beautiful way

Discussion: you discuss in a concise but exhaustive way the results and the consequence of your findings.

Conclusion are fine

Author Response

Thank you.
